# SARS-CoV-2 Gamma and Delta Variants of Concern Might Undermine Neutralizing Activity Generated in Response to BNT162b2 mRNA Vaccination

**DOI:** 10.3390/v14040814

**Published:** 2022-04-15

**Authors:** Luigia Trabace, Lorenzo Pace, Maria Grazia Morgese, Isabel Bianca Santo, Domenico Galante, Stefania Schiavone, Dora Cipolletta, Anna Maria Rosa, Pierluigi Reveglia, Antonio Parisi, Paolo Tucci, Giovanni Pepe, Rodolfo Sacco, Maria Pia Foschino Barbaro, Gaetano Corso, Antonio Fasanella

**Affiliations:** 1Department of Clinical and Experimental Medicine, University of Foggia, 71122 Foggia, Italy; mariagrazia.morgese@unifg.it (M.G.M.); stefania.schiavone@unifg.it (S.S.); pierluigi.reveglia@unifg.it (P.R.); paolo.tucci@unifg.it (P.T.); gaetano.corso@unifg.it (G.C.); 2Istituto Zooprofilattico Sperimentale della Puglia e della Basilicata, 71121 Foggia, Italy; lorenzo.pace@izspb.it (L.P.); domenico.galante@izspb.it (D.G.); dora.cipolletta@izspb.it (D.C.); antonio.parisi@izspb.it (A.P.); antonio.fasanella@izspb.it (A.F.); 3Laboratory of Clinical Pathology, Policlinico Riuniti di Foggia, 71122 Foggia, Italy; isabel.bsanto@gmail.com (I.B.S.); arosa@opsedaliriunitifogga.it (A.M.R.); 4Centro Antiveleni, Policlinico Riuniti di Foggia, 71122 Foggia, Italy; pepegiovannilucera@tiscali.it; 5Gastroenterology Unit, Department of Medical Sciences, Policlinico Riuniti di Foggia, 71122 Foggia, Italy; saccorodolfo@hotmail.com; 6Department of Medical and Surgical Sciences, Institute of Respiratory Diseases, University of Foggia, 71122 Foggia, Italy; mariapia.foschino@unifg.it

**Keywords:** BNT162b2 mRNA, SARS-CoV-2 VOCs, neutralizing antibodies

## Abstract

The Delta variant raised concern regarding its ability to evade SARS-CoV-2 vaccines. We evaluated a serum neutralizing response of 172 Italian healthcare workers, three months after complete Comirnaty (BNT162b2 mRNA, BioNTech-Pfizer) vaccination, testing their sera against viral isolates of Alpha, Gamma and Delta variants, including 36 subjects with a previous SARS-CoV-2 infection. We assessed whether IgG anti-spike TRIM levels and serum neutralizing activity by seroneutralization assay were associated. Concerning Gamma variant, a two-fold reduction in neutralizing titres compared to the Alpha variant was observed, while a four-fold reduction of Delta virus compared to Alpha was found. A gender difference was observed in neutralizing titres only for the Gamma variant. The serum samples of 36 previously infected SARS-CoV-2 individuals neutralized Alpha, Gamma and Delta variants, demonstrating respectively a nearly three-fold and a five-fold reduction in neutralizing titres compared to Alpha variant. IgG anti-spike TRIM levels were positively correlated with serum neutralizing titres against the three variants. The Comirnaty vaccine provides sustained neutralizing antibody activity towards the Alpha variant, but it is less effective against Gamma and even less against Delta variants.

## 1. Introduction

Through genomic surveillance of the severe acute respiratory syndrome-related coronavirus 2 (SARS-CoV-2), in addition to the original virus first identified in Wuhan, China, a number of new variants have been identified in recent months [1]. In this regard, the World Health Organization prompted the characterization of specific Variants of Interest (VOIs) and Variants of Concern (VOCs) [2]. At the present, several variants of concern emerged that demonstrate increased transmissibility and/or evasion of immune responses from prior SARS-CoV-2 infection [3].

A distinct phylogenetic cluster (named Alpha variant, Pango lineage B.1.1.7 or GISAID 501Y.V1) was first detected in September 2020 in England. This strain has now spread to more than 50 countries, and this has resulted in increased hospitalizations, overstretched health systems and excess mortality.

Furthermore, a Beta variant (Pango lineage B.1.351 or GISAID 501Y.V2) was characterized in South Africa in May 2020. This variant is also associated with increased transmissibility, and it has also been increasingly reported in European Union/European Economic Area (EU/EEA) countries. Although reported at lower levels, it is of equal concern that another variant known as Gamma (Pango lineage P.1 or GISAID 501Y.V3) is increasing rapidly in Brazil since November 2020 and spreading far beyond.

The Delta variant (Pango lineage B.1.617.2 or GISAID 478K.V1), first detected in India during October 2020, has overtaken the Alpha variant [1,4].

All analyses on secondary attack rates and household transmission data continue to support increased transmissibility. There is in vitro evidence suggestive of increased replication in biological systems that model human airways [5]. Evidence from England and Scotland suggests that there may be an increased risk of hospitalization compared to contemporaneous Alpha cases [6,7]. Evidence accumulated since the first threat assessment brief on the emergence of the SARS-CoV-2 Delta variant in India, published on May 2021, resulted in the Delta variant being upgraded from a VOI to a VOC. Based on the available evidence, the SARS-CoV-2 Delta VOC is 40–60% more transmissible than the Alpha VOC and may be associated with a higher risk of hospitalization [8].

In November 2021, a new variant in South Africa, named the Omicron variant, was detected and reported to the WHO. The Omicron variant is the most heavily mutated variant among all the VOC so far, paving the way for enhanced transmissibility [9].

During December 2020, the Comirnaty (BNT162b2 mRNA, BioNTech-Pfizer) vaccine was shown to be 95% efficacious in preventing symptomatic COVID-19. Real-world data demonstrated 95% effectiveness of this mRNA-based vaccine against the original SARS-CoV-2 and Alpha variant [10,11].

In the present study, we describe the neutralizing response against viral isolates of the Alpha, Gamma and Delta VOCs of sera from Italian healthcare workers, collected three months after the second BNT162b2 vaccine dose. Moreover, we evaluated the neutralizing activity of both doses of the vaccine in previously infected SARS-CoV-2 individuals. Finally, we investigated the association between serum IgG anti-spike TRIM levels and serum neutralizing activity against Alpha, Gamma and Delta variants by seroneutralization assay.

## 2. Materials and Methods

### 2.1. Subjects

Data from 172 vaccinated volunteers (36 of them had been infected with SARS-CoV-2 before vaccination) were included in this paper [72 males, mean age 48.54 years (95% CI = 47–55); 100 females mean age 40.96 years (95% CI = 36–45)]. Neutralizing antibodies against Alpha, Gamma and Delta VOCs were tested in health care workers (Azienda Ospedaliero-Universitaria of Foggia, Italy), 3 months after having received the second Comirnaty dose.

### 2.2. Cells and Virus Stock

A seroneutralization test was performed in a biosafety level 3 laboratory at the Istituto Zooprofilattico Sperimentale della Puglia e della Basilicata (Foggia, Italy). Three variants of the virus SARS-CoV-2, Alpha (GSAID ID: EPI_ISL_745193), Gamma (GSAID ID: EPI_ISL_1819245) and Delta (GSAID ID: EPI_ISL_1919755), were isolated by our group and used for this aim. African green monkey kidney Vero E6 cells were used for both propagations of SARS-CoV-2 and neutralization assay. Cells were cultured into a 25 cm^2^ cell culture flask in Eagle’s minimal essential medium (EMEM) [12].

### 2.3. Titration of SARS-CoV-2

Virus infectious titres were established by the Reed and Muench tissue culture infective dose (TCID50) end point method [13]. For titration, 2 × 10^4^ Vero E6 cells (in 50 µL) were plated into 96-well plates. Stock solution of SARS-CoV-2 was diluted serially from 10^−1^ to 10^−8^, and 25 µL of each dilution were added to cells and incubated in 5% CO_2_ at 37 °C for 72 h. Ten replicates were performed for each dilution and used to quantify the virus titre and TCID50 determination.

### 2.4. Cytopathic Effect Based Micro-Neutralization Assay

A cytopathic effect-based micro-neutralization assay was conducted in 96-well plates. Eight-fold dilutions (1:20, 1:40, 1:80, 1:160, 1:320, 1:640, 1:1280, 1:2560) of serum samples were tested in triplicate wells for the presence of antibodies that neutralized the infectivity of three SARS-CoV-2 variants (Alpha, Gamma and Delta) in Vero E6 cell monolayers. In addition, 100 TCID50 of the virus in 25 µL/well were incubated with 25 µL of each dilution of serum in EMEM with 6% FBS for 1 h at 37 °C. After incubation, 2 × 10^4^ Vero E6 cells (in 50 µL) were added to each well. Neutralizing antibody titre was defined as the last serum dilution at which no cytopathic effect breakthrough was observed [14,15].

### 2.5. Anti-Spike-IgG Measurements

A chemiluminescent immunoassay to analyze the SARS-CoV-2-specific antibodies IgG (IgG anti-spike TRIM) [16] against three main domains of SARS-CoV-2 protein (SARS-CoV-2 TrimericS IgG—Diasorin, Italy) was used. Precision of the method for IgG determinations was evaluated by measuring daily internal quality controls (IQCs). The first IQC was a negative sample with a level less than 1.85 AU/mL; it showed a negativity agreement of more than 98% (n = 50), and a positive control with an average of 36.41 AU/mL with a variability (CV%) of 9.2% (n = 50). Moreover, we prepared two human serum pools from our routine samples with average levels of IgG of 3.25 AU/mL (CV%= 8.9%, n = 50) and of 37.85 AU/mL (CV% = 9.8, n = 50).

### 2.6. Statistical Analysis

GraphPad software, version 9.0, was used. In two of the tested samples, the neutralizing titre was found under the lower dilution; therefore, statistical evaluation was performed on 170 subjects. To assess differences in the neutralizing antibody titres against Alpha, Gamma and Delta variants, as well as in the ratio between serum IgG anti-spike TRIM levels and serum neutralizing titres, Kruskal–Wallis or Mann–Whitney tests were performed. Differences in the percentage of mean decrease relative to Alpha variant were verified by one-way ANOVA, followed by Tukey’s post hoc test. To evaluate whether IgG anti-spike TRIM levels were correlated to serum neutralizing activity against Alpha, Gamma and Delta variants, linear regression analysis was performed. Results were expressed as geometric mean titres (GMT) with lower and upper 95% confidence intervals (CI) or mean ± standard error of mean (SEM).

## 3. Results

### 3.1. Serum Samples Neutralizing Activity against Alpha, Gamma and Delta Virus Isolates

We tested neutralizing antibodies against Alpha, Gamma and Delta variants using serum samples obtained from healthcare workers who had received two doses of BNT162b2 vaccine, collected three months after the second dose (n = 134, with no COVID-19 diagnosis). As shown in Figure 1, serum samples had neutralizing activity against Alfa, Gamma and Delta virus isolates, with geometric mean titres (GMT) of 191.1 (95% CI = 159.6–229.0), 82.38 (95% CI = 71.00–95.60) and 47.45 (95% CI = 40.96–54.96), respectively. Results related to the Gamma variant demonstrated a two-fold reduction in neutralizing titres compared with the Alpha variant in vaccinated individuals while neutralizing antibodies were four-fold reduced against the Delta virus relative to Alpha (Figure 1 inset, one-way ANOVA F (2-402) = 15.34, followed by Tukey’s post hoc test Alpha vs. Gamma *p* < 0.0001 and Alpha vs. Delta *p* < 0.0001, Gamma vs. Delta *p* = 0.5482). Statistical analysis demonstrated significant differences in neutralizing titres among Alpha, Gamma and Delta variants (Figure 1, Kruskal–Wallis statistic = 109.08 followed by Dunn’s multiple comparisons test among groups *p* < 0.0001). From a gender analysis, we found a significant difference between sexes in neutralizing titres against the Gamma variant with higher GMT values in females with respect to males (Figure 2b, GMT = 69.39, 95% CI = 55.93–86.08 for male and 94.48, 95% CI = 77.27–115.5 for female, Mann–Whitney test, *p* = 0.024). Considering the significant difference in mean age between male and female subjects (Student’s *t* test *p* = 0.0003), we correlated neutralizing titres against Gamma variant with age. We found that neutralizing titres against the Gamma variant were significantly and negatively correlated with age in men (Pearson correlation R2 = 0.3, *p* = 0.0199), but not in women (R2 = 0.03, *p* = 0.814). However, no differences were observed in serum samples neutralizing activity against Alpha variants between males and females (Figure 2a, GMT = 166.9 for male, 95% CI = 127.7–218.2 and 180.5 for female, 95% CI = 145.0–224.8, Mann–Whitney test, *p* = 0.8227) and Delta (Figure 2c, GMT = 43.87 for male, 95% CI = 35.21–54.66 and 49.93 for female, 95% CI = 40.88–60.99, Mann–Whitney test, *p* = 0.35).

### 3.2. Serum Samples Neutralizing Activity against Alpha, Gamma and Delta Virus Isolates in Previously Infected SARS-CoV-2 Individuals

In previously infected SARS-CoV-2 individuals (n = 36), serum samples neutralized Alpha, Gamma and Delta virus isolates with GMT of 1437 (95% CI = 1060–1948), 527.9 (95% CI = 379.6–743.2) and 290.6 (95% CI = 213.6–395.4), respectively (Figure 3, Kruskal–Wallis statistic = 45.41 followed by Dunn’s multiple comparisons test Alpha vs. Delta *p* < 0.0001, Alpha vs. Gamma *p* < 0.0001 and Gamma vs. Delta *p* = 0.0567), demonstrating a nearly three-fold and a five-fold reduction in neutralizing titres compared with Alpha variant (Figure 3 inset, one-way ANOVA F(2-108) = 50.91, followed by Tukey’s post hoc test Alpha vs. Gamma and Delta *p* < 0.0001 and Gamma vs. Delta *p* = 0.1076).

### 3.3. Correlation between Serum IgG Anti-Spike TRIM Levels and Serum Neutralizing Titres against Alpha, Gamma and Delta Variants

Linear regression analyses revealed that serum IgG anti-spike TRIM levels were positively correlated with serum neutralizing titres against Alpha, Gamma and Delta variants (Figure 4a–c, Alpha R2 = 0.6004, Gamma R2 = 0.8075, Delta R2 = 0.7589). In order to obtain a factor that includes the weight of both measures, we calculated the ratio between the titre of neutralizing antibodies with respect to the level of total IgG anti-spike TRIM. The ratios between diluting neutralizing titres serum and the IgG anti-spike TRIM levels highlighted that statistically significant differences exist among the three variants (Figure 5, Kruskal–Wallis statistic = 226.7 followed by multiple comparisons test Alpha vs. Delta *p* < 0.0001, Alpha vs. Gamma and Gamma vs. Delta *p* < 0.0001).

## 4. Discussion

The emergence of new VOCs of SARS-CoV-2 urgently requires active research to understand their phenotypic and antigenic properties of relevance for the pandemic and its control. This study contributes to the gathering of information on the neutralizing antibody activity of BNT162b2 fully vaccinated individuals towards three variants of SARS-CoV-2 in Italy. Our neutralization experiments suggested that this vaccine seems to be less effective against Delta and Gamma mutations in a setting of a young adult population, three months after the second dose, compared to the Alpha variant.

Moreover, we found, although in a small sample, that these results are confirmed also in previously infected fully vaccinated individuals. COVID-19 naive participants showed lower neutralizing antibody titres compared to COVID-19 recovered and vaccinated subjects [17]. These results are of particular interest if we consider that available vaccines anti-SARS-CoV-2 stimulate systemic immune responses to spike proteins, whereas natural infection also induces immune responses against the several other open reading frames encoded by the nearly 29,900 nucleotides of the virus [18]. Although no virus genotyping was effectively carried out in these patients, available data regarding the mapping of variant diffusion in Italy and in the Apulia region at the time of infection (November 2020–March 2021) revealed that the predominant variant was the Alpha (Italy 73.06% and Apulia 98.3%); thus, we can assume that this cluster of subjects could have been previously infected by the Alpha variant [19]. However, no seroconversion data were available in the uninfected population before vaccination; hence, although all of them reported to be COVID-19 naïve, we cannot completely rule out the possibility of previous asymptomatic infections.

We observed lower serum neutralization efficiency of the Gamma and Delta VOCs compared to the Alpha variant, and only two sera of vaccinated subjects in our setting showed a complete escape of the examined variants, suggesting that 98.84% of the tested population was able to produce detectable levels of neutralizing antibodies. The SARS-CoV-2 variant B.1.617, as well as the P.1 (all containing a mutation at position E484) display increased transmissibility both due to their higher affinity for the cell receptor ACE2 and these features, as well as the remodeling of the N-terminal domain, could explain their ability to partially bypass immunity generated after vaccination [20,21].

Our data are in line with recent results showing that two doses of BNT162b2 vaccine elicited reduced neutralizing antibody activity against Delta variant relative to wild-type SARS-CoV-2 first detected in Wuhan, China, and significantly more reduced than against the Alpha variant [22]. However, a recent test-negative case control study demonstrated that modest differences were retrieved after two BNT162b2 vaccine doses in terms of symptomatic cases positive to Alpha, Gamma and Delta VOCs [23].

Interestingly, in our study serum IgG levels of BNT162b2, fully vaccinated subjects positively correlated with neutralizing antibody titres for all the three examined variants. In this regard, although recently many studies have highlighted how the mutations present in the S proteins of VOCs impact host cell interactions and antibody-mediated neutralization [24,25,26], our results suggested that serum IgG measurements could be used as an easier and reliable approach to identify a reduced neutralizing capacity of sera, regardless of VOCs considered. Thus, since in our study neutralization strongly and directly correlated with the abundance of antibodies against the three main domains of SARS-CoV-2 protein (receptor-binding domain of coronavirus’s S protein, S1 and S2), the chemiluminescent immunoassay used here could serve as a rapid and convenient screening method for identifying neutralizing antibody activity [27]. The relevance of these data is particularly noticeable if we consider the laborious nature of cytopathic effect measuring, which requires a high biosafety containment level, that makes it an inappropriate tool for testing large sample sets and for surveillance purposes. Moreover, we considered the ratio between the titre of neutralizing antibodies with respect to the level of total anti-S IgG. This factor has been taken into account because of the positive correlation between anti-S IgG (TRIM) and the neutralizing titre. The value of this ratio greater than 1 indicated that the neutralizing serum dilution is greater than the total IgG expressed as arbitrary units (AU/mL). Therefore, the higher this ratio, the greater the neutralizing power of the patient’s serum. Thus, the identification of this value further confirmed that significant differences exist among the three variants. Remarkably, it is now known that COVID-19 may exhibit gender disparities, as already demonstrated for influenza A virus infections [28]. It has been shown that men account for higher hospitalizations and deaths. However, notwithstanding clear gender differences in COVID-19 outcomes, few and conflicting reports on gender-targeted research are currently available [29,30].

In this scenario, our data showed that only the Gamma VOC appears to be less efficiently inhibited by sera from vaccinated males, compared to females. Thus, at least based on the available results, the grade of susceptibility of Gamma VOC appears to be gender-dependent. Indeed, when data were disaggregated by age, we found that neutralizing antibody titre was negatively correlated with age only in the male population, while, in females, no significant difference was evident.

## 5. Conclusions

In conclusion, although implications of waning antibody levels are not yet clear, our results support the notion that the mRNA BNT162b2 vaccine provides neutralizing activity towards the different strains, especially against Alpha VOC, but it is less effective against Gamma and even less against Delta variants. Remarkably, our results demonstrated that the emerging Delta variant partially, although notably, escapes neutralizing elicited by vaccination or previous infection with other variants, highlighting the need for control measures to prevent variants from further spreading, including the design of next-generation vaccines, the use of alternative viral antigens, or an additional vaccine dose. This study can represent a successful model to use also for further future studies on the evaluation antibody neutralizing activity of vaccines against the current circulating variants Omicron (BA.1) and Omicron 2 (BA.2).

## Figures and Tables

**Figure 1 viruses-14-00814-f001:**
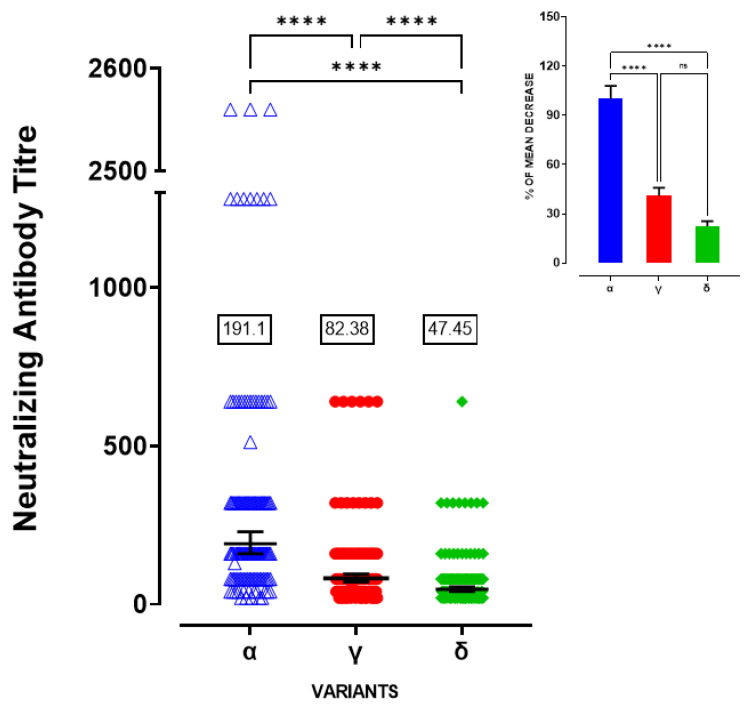
Neutralizing antibody titre against Alpha, Gamma and Delta variants **** *p* < 0.0001; Inset: Percentage (%) of mean decrease for neutralizing antibody titre against Gamma and Delta variants with respect to Alpha **** *p* < 0.0001; ns = not significant.

**Figure 2 viruses-14-00814-f002:**
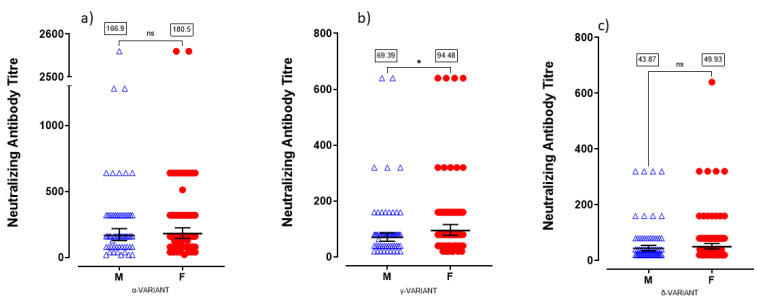
Gender effect on neutralizing antibody titre against variants of concern: (**a**) neutralizing antibody titre against Alpha variant in male and female subjects, ns = not significant; (**b**) neutralizing antibody titre against Gamma variant in male and female subjects, * *p* < 0.05; (**c**) neutralizing antibody titre against Delta variant in male and female subjects, ns = not significant.

**Figure 3 viruses-14-00814-f003:**
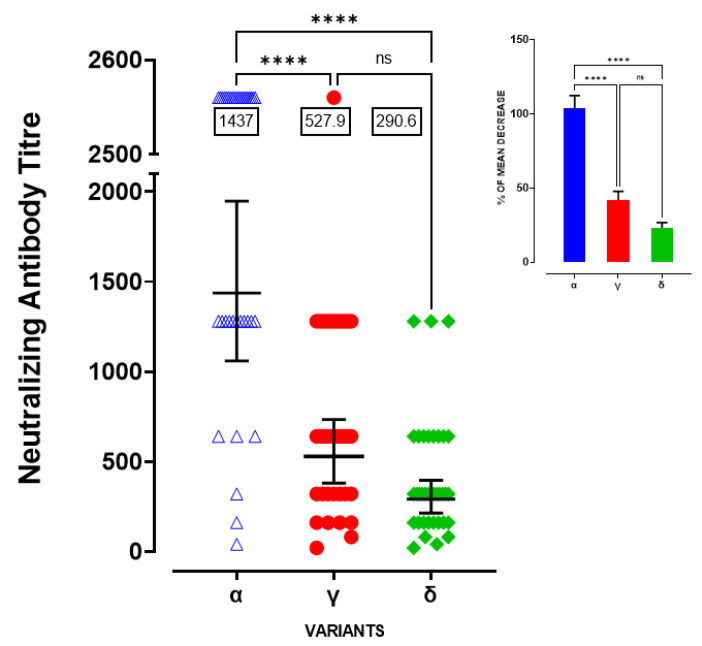
Neutralizing antibody titre against against variants of concern in COVID-19 recovered subjects. Neutralizing antibody titre against Alpha, Gamma and Delta variants in COVID-19 recovered subjects, **** *p* < 0.0001, ns = not significant; Inset: Percentage (%) of mean decrease for neutralizing antibody titre against Gamma and Delta variants with respect to Alpha **** *p* < 0.0001, ns = not significant.

**Figure 4 viruses-14-00814-f004:**
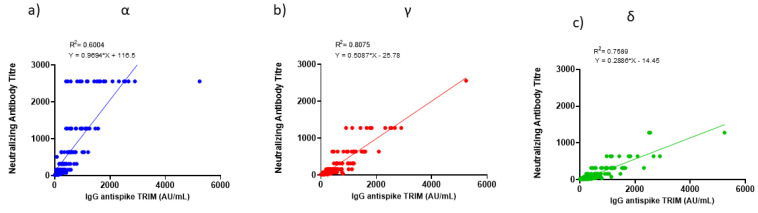
Correlation between neutralizing antibody titres against variants of interest and IgG anti-spike TRIM levels (**a**) correlation between neutralizing antibody titres against Alpha variants and IgG anti-spike TRIM levels; (**b**) correlation between neutralizing antibody titres against Gamma variants and IgG anti-spike TRIM levels; (**c**) correlation between neutralizing antibody titres against Delta variants and IgG anti-spike TRIM levels.

**Figure 5 viruses-14-00814-f005:**
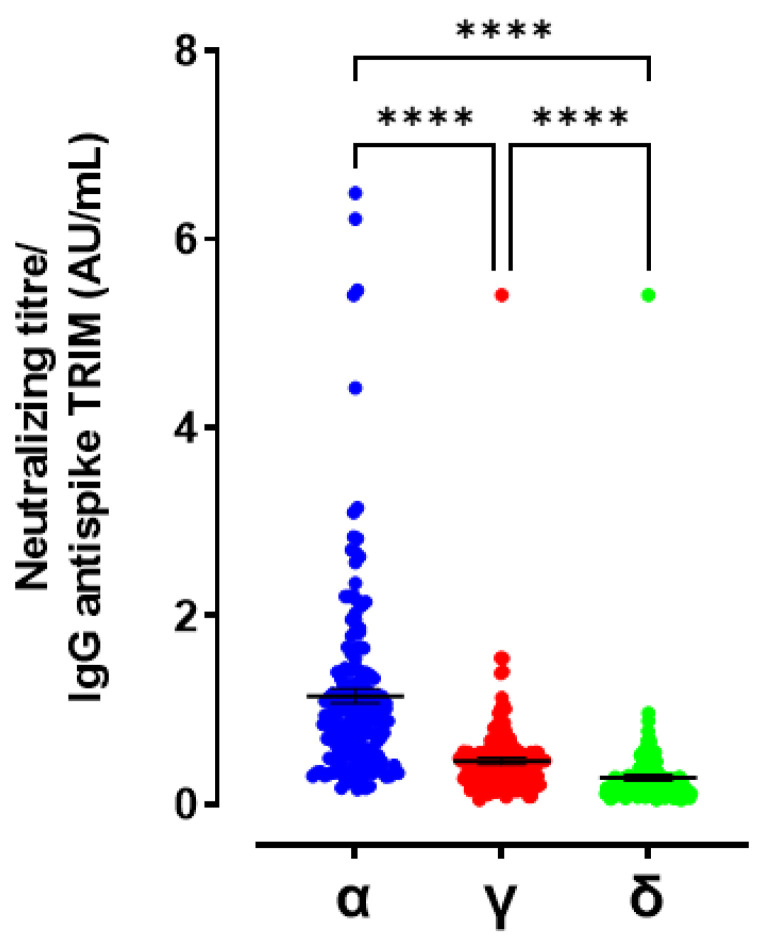
Ratio between neutralizing antibody titre against Alpha, Gamma and Delta variants and IgG anti-spike TRIM levels **** *p* < 0.0001.

## Data Availability

The data presented in this study are available on request from the corresponding author.

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
