# Peer review of "SARS-CoV-2 Gamma and Delta Variants of Concern Might Undermine Neutralizing Activity Generated in Response to BNT162b2 mRNA Vaccination"

_viruses, 2022, doi:10.3390/v14040814_

Round 1

Reviewer 1 Report

  1. Figure 2: There is a difference on medium age between male and female. The statistics should be corrected by taking in the age factor.
  2. Figure 1: The inset figure is too small.
  3. The subjects who were previously infected showed much higher neutralization end point titers to variants. Can more information be provided? For example, how long ago they were infected with what strain. If no information on strain, the time frame may help to estimate the exposure as the variants transmitted at different time. Some info of the transmission of different variants should also be in the intro or in the discussion.  Although these variants are now all replaced by new variants, the overall value of the manuscript is still good since they reflect immune escape of the strains bearing a unique set of mutations. 
  4. Some discussion on the variants in terms of mutations, affinity to ACE-2, and etc., can help explaining why delta had a greatest escape as compared to the other two variants.
  5. At the time of sample collection, has there any documentation that all three strains or some of them already circulated in the area? Since the study is a cross-section study, there is no baseline data. The difference between variants in terms of immune response could be affected by silent sero-conversion. Could the authors comment on this? Has any sero-conversion been measured in the uninfected population to ensure they were truely naive before vaccination? 
  6. Page 1, line 44-45, rephrase
  7. Line 92: Sieroneutralization: spell check

Author Response

Dear Reviewer,

thank you for your valuable comments and for helping us to improve our manuscript.

  • Figure 2: There is a difference on medium age between male and female. The statistics should be corrected by taking in the age factor.

Answer: Thanks for your comment. Indeed, a significant difference was found between mean age of male and female subjects. As regard to neutralizing antibody titre against gamma variant, we performed Pearson correlation in order to taking into account age factor.

  • Figure 1: The inset figure is too small.

Answer: Thanks for your comment. We have enlarged all figures

  • The subjects who were previously infected showed much higher neutralization end point titers to variants. Can more information be provided? For example, how long ago they were infected with what strain. If no information on strain, the time frame may help to estimate the exposure as the variants transmitted at different time. Some info of the transmission of different variants should also be in the intro or in the discussion.  Although these variants are now all replaced by new variants, the overall value of the manuscript is still good since they reflect immune escape of the strains bearing a unique set of mutations. 

Answer: Thanks for your comment. Although no virus genotyping was effectively carried out in these patients, available data regarding the mapping of variant diffusion in Italy and in Apulia region at the time of infection (November 2020- march 2021) revealed that the predominant variant was the alpha (Italy 73.08% and Apulia 98.3%), thus we can assume that this cluster of subjects could have been previously infected by the alpha variant.

  • Some discussion on the variants in terms of mutations, affinity to ACE-2, and etc., can help explaining why delta had a greatest escape as compared to the other two variants.

Answer: Thanks for your precise comment. We have added in the discussions what you asked

  • At the time of sample collection, has there any documentation that all three strains or some of them already circulated in the area? Since the study is a cross-section study, there is no baseline data. The difference between variants in terms of immune response could be affected by silent sero-conversion. Could the authors comment on this? Has any sero-conversion been measured in the uninfected population to ensure they were truely naive before vaccination? 

Answer: Thanks for your comment. Unfortunately, no sero-conversion data are available in the uninfected population before vaccination, so cannot exclude the possibility of a previous asymptomatic infection. This limitation was added in the discussion.

  • Page 1, line 44-45, rephrase

Answer: Done

  • Line 92: Sieroneutralization: spell check

Answer: Done

Reviewer 2 Report

Trabace et al. completed an interesting study aimed to determine, in subjects who received two doses of BTN162b2 mRNA vaccine, association between IgG anti-spike  TRIM levels and serum neutralizing activity against alpha, gamma and delta virus variants. The study is conducted at a high-quality standard and the paper is well written, the methods are appropriate and the results are clearly reported and deeply discussed. It is an important contribution to the literature on this topic and provides a useful model to analyze vaccine response against future SARS-Cov-2 variants.

I have just a suggestion for the discussion section. I think that it would be interesting to analyze more thoroughly the phenomenon of the gender difference in vaccine-induced response (see Morgan R. Curr Opin Virol 2019;35:35-41; Vassallo A. Front Glob Women’s Health 2021;2:761511). Also, in order to support the role of the serum IgG level as a reliable approach to identify subjects who may not develop a viral neutralizing titer additional references should be added (see Papadopoli R. Vaccines, 2021;9(12):1494).   

Author Response

Dear Reviewer,

thank you for your valuable comments and for helping us to improve our manuscript.

I have just a suggestion for the discussion section. I think that it would be interesting to analyze more thoroughly the phenomenon of the gender difference in vaccine-induced response (see Morgan R. Curr Opin Virol 2019;35:35-41; Vassallo A. Front Glob Women’s Health 2021;2:761511). Also, in order to support the role of the serum IgG level as a reliable approach to identify subjects who may not develop a viral neutralizing titer additional references should be added (see Papadopoli R. Vaccines, 2021;9(12):1494).   

Answer: Thanks for the appropriate comment. Our group fully agrees with you and we have added the references that you suggested.

Reviewer 3 Report

Luigia Trabace et al. collected the serums from 172 local healthcare workers after complete vaccination, and mainly tested their neutralizing activity against three VOCs, alpha, gamma, and delta variants. And they find that neutralization activities against gamma and delta variants were both weaker than that against alpha variant. While the immune evasion of the VOCs against serum from vaccinated people or these recovered from COVID-19 has been extensively studied. The novelty of this study is limited. Hope the suggestions could be helpful.

For comparison of the neutralizing activities of the VOCs, it will be good to have the WA1 strain or the WA1 plus D614G mutation as a control.

It’s interesting that the authors found a significant difference between sexes in neutralizing titers against gamma variant with higher GMT values in females with respect to males. It will be good to have some references to support the results. There is also study show the opposite, https://journals.asm.org/doi/10.1128/mSphere.00275-21

In the experiments to evaluate the correlation between serum spike binding ability and serum neutralizing ability, which spike was used? From WA1 or the variant?

Line 237 this statement seems not accurate. Many studies were aiming at explaining the mechanism of the influences of the spike mutations on the receptor binding and antibody escape. Wanwisa Dejnirattisai et al. Antibody evasion by the P.1 strain of SARS-CoV-2. Cell. 2021

Pengfei Wang et al. Antibody Resistance of SARS-CoV-2 Variants B.1.351 and B.1.1.7. Nature. 2021

Katherine G. Nabel et al. Structural basis for continued antibody evasion by the SARS-CoV-2 receptor binding domain. Science. 2021

Yiska Weisblum et al. Escape from neutralizing antibodies by SARS-CoV-2 spike protein variants. eLife. 2020

Wilfredo F. Garcia-Beltran et al. Multiple SARS-CoV-2 variants escape neutralization by vaccine-induced humoral immunity. Cell. 2021

Rita E. Chen et al. Resistance of SARS-CoV-2 variants to neutralization by monoclonal and serum-derived polyclonal antibodies. Nature medicine. 2021

Meng Yuan et al. Structural and functional ramifications of antigenic drift in recent SARS-CoV-2 variants. Science. 2021

Qianqian Li et al. The Impact of Mutations in SARS-CoV-2 Spike on Viral Infectivity and Antigenicity. Cell. 2020

Line 255-256 Please include the references.

Author Response

Dear Reviewer,

thank you for your valuable comments and for helping us to improve our manuscript.

  • For comparison of the neutralizing activities of the VOCs, it will be good to have the WA1 strain or the WA1 plus D614G mutation as a control.

Answer: Thank you for your appropriate comment. As previously described by our group [Rondinone et al, Viruses, 2021], the VOC202012/01 variant is sensitive to the neutralizing activity of antibodies produced by patients in response to previously circulating viral strains, and the neutralizing titers are identical to those established by using the lineage B.1 with the D614G mutation. For these reason in this paper we have decided to use alpha variant as control

  • It’s interesting that the authors found a significant difference between sexes in neutralizing titers against gamma variant with higher GMT values in females with respect to males. It will be good to have some references to support the results. There is also study show the opposite, https://journals.asm.org/doi/10.1128/mSphere.00275-21

Answer: Thanks for the comment. We have added the references that you suggested.

  • In the experiments to evaluate the correlation between serum spike binding ability and serum neutralizing ability, which spike was used? From WA1 or the variant?

Answer: The correlation reported in Fig.4, is related to serum levels of IgG anti-spike AU/ml (method TRIM Diasorin) against serum neutralizing titres in patients sera.

  • Line 237 this statement seems not accurate. Many studies were aiming at explaining the mechanism of the influences of the spike mutations on the receptor binding and antibody escape. Wanwisa Dejnirattisai et al. Antibody evasion by the P.1 strain of SARS-CoV-2. 2021

Pengfei Wang et al. Antibody Resistance of SARS-CoV-2 Variants B.1.351 and B.1.1.7. Nature. 2021

Katherine G. Nabel et al. Structural basis for continued antibody evasion by the SARS-CoV-2 receptor binding domain. Science. 2021

Yiska Weisblum et al. Escape from neutralizing antibodies by SARS-CoV-2 spike protein variants. eLife. 2020

Wilfredo F. Garcia-Beltran et al. Multiple SARS-CoV-2 variants escape neutralization by vaccine-induced humoral immunity. Cell. 2021

Rita E. Chen et al. Resistance of SARS-CoV-2 variants to neutralization by monoclonal and serum-derived polyclonal antibodies. Nature medicine. 2021

Meng Yuan et al. Structural and functional ramifications of antigenic drift in recent SARS-CoV-2 variants. Science. 2021

Qianqian Li et al. The Impact of Mutations in SARS-CoV-2 Spike on Viral Infectivity and Antigenicity. Cell. 2020

Answer: Thanks for your precise comment. We have rephrased the sentence in the discussions and added recommended references

  • Line 255-256 Please include the references.

Answer: Done

Round 2

Reviewer 3 Report

My questions have been addressed.